# What is a Number, That a Large Language Model May Know It?

**Raja Marjieh**[*]                                                                *raja.marjieh@princeton.edu*
*Department of Psychology*
*Princeton University*

**Veniamin Veselovsky**[*]                                                          *veniamin@princeton.edu*
*Department of Computer Science*
*Princeton University*

**Thomas L. Griffiths**[†]                                                          *tomg@princeton.edu*
*Departments of Psychology and Computer Science*
*Princeton University*

**Ilia Sucholutsky**[†]                                                            *is3060@nyu.edu*
*Center for Data Science*
*New York University*

[*/†]*Equal contribution.*

**Reviewed on OpenReview:** *https://openreview.net/forum?id=dnXulFHKbw*

## Abstract

Numbers are a basic part of how humans represent and describe the world around them. As a consequence, learning effective representations of numbers is critical for the success of large language models as they become more integrated into everyday decisions. However, these models face a challenge: depending on context, the same sequence of digit tokens, e.g., 911, can be treated as a number or as a string. What kind of representations arise from this duality, and what are its downstream implications? Using a similarity-based prompting technique from cognitive science, we show that LLMs learn representational spaces that blend string-like and numerical representations. In particular, we show that elicited similarity judgments from these models over integer pairs can be captured by a combination of Levenshtein edit distance and numerical Log-Linear distance, suggesting an entangled representation. In a series of experiments we show how this entanglement is reflected in the latent embeddings, how it can be reduced but not entirely eliminated by context, and how it can propagate into a realistic decision scenario. These results shed light on a representational tension in transformer models that must learn what a number is from text input.

## 1 Introduction

> "What is a number, that a man may know it, and a man, that he may know a number?"
>
> *Warren McCulloch* (1961)

Numbers play a pivotal role in many aspects of human cognition (Dehaene, 2011). Starting from a narrow sense of magnitude in infancy, humans gradually acquire nuanced representations of numbers that capture abstract properties such as "even" and "odd" (Miller & Gelman, 1983; Piantadosi et al., 2014). Decades

of research have been devoted to understanding how humans process and represent numbers (Cheyette & Piantadosi, 2020; Miller & Gelman, 1983; Nieder & Dehaene, 2009; Piantadosi, 2016; Tenenbaum, 1999). In fact, this endeavor dates back to some of the earliest applications of neural networks to artificial intelligence (McCulloch, 1961; McCulloch & Pitts, 1943) when researchers sought to define computing machines (a simulated "man") that can intelligently process numbers.

The question of how computing machines represent numbers assumes key importance as modern neural networks, in particular large language models (LLMs), are becoming integrated into everyday decisions (Collins et al., 2024; 2025; Hanna et al., 2023; McCoy et al., 2024a;b; Oktar et al., 2026; Stolfo et al., 2023; Tian et al., 2023; Zhu et al., 2025). Since these models learn representations by predicting tokens in text, LLMs face a challenge: depending on context, the same sequence of digit tokens, e.g., 101 or 911, can be treated as a number or as a string. This duality introduces an issue akin to polysemy and homonymy (Vicente & Falkum, 2017), but it extends across different symbolic systems. What kind of representations arise from this duality, and what are its downstream implications? This is not straightforward to answer, in part because LLMs lack the experience that shapes humans' number representations, and in part because model internals are not always available and diagnostic tests can be challenging to design.

Here we address this challenge by leveraging tools from cognitive science for characterizing representations (Marjieh et al., 2024a; Shepard, 1980; 1987; Tenenbaum & Griffiths, 2001; Tversky & Hutchinson, 1986). Specifically, we elicit similarity judgments across number pairs from six modern LLMs using an appropriate prompt (see also Appendix F.4 for an additional model). We then use those judgments to construct detailed maps (similarity matrices) that capture how the models organize numbers. Crucially, this technique can applied to any model without the need to access its internals. Moreover, when the model internals are available, this approach can be combined with probing techniques to evaluate how prompt behavior is reflected in the structure of the latent embeddings.

We show that despite variation in model training data and size, all models exhibit highly regular patterns that can be adequately decomposed in terms of a psychologically-motivated numerical distance based on a Log-Linear representation of numbers (Piantadosi, 2016) and a string edit distance. This suggests that these representations may be entangled within the models. We show how this entanglement can be reduced but not eliminated by introducing a context that specifies the 'type' of the integer (`int()` vs. `str()`), and how it can be probed internally in a model's latent space. Inspired by these results we then construct a decision scenario that reveals how string bias can propagate into a realistic setting leading to incorrect answers. Viewed together, these findings shed light on an intrinsic tension between number and string representations that modern large language models must learn to navigate.

To summarize, the contributions of our work are as follows:

- We introduce a general technique from cognitive science based on similarity judgments for characterizing the representations of LLMs without having to access their internals.

- We apply this technique to the domain of numbers, revealing representations that blend numerical and string components across six different models.

- We show how similarity judgments can be combined with standard probing techniques to shed light on how prompt behavior is also reflected in model latents.

- We show how this approach can be used to motivate additional experiments that reveal decision errors related to number understanding in LLMs.

## 2 Related Work

### 2.1 Numbers and Language Models

The processing and representation of numbers in large language models is a topic of active research. Number representations have recently been studied within pretrained language models using techniques from mechanistic interpretability. For example, Zhu et al. (2025) devised a dataset of addition problems to show that

LLMs encode the value of numbers linearly in that context, whereas Levy & Geva (2024) showed that LLMs represent each digit separately in base 10. Likewise, circuit analysis has been adopted to characterize how smaller language models compute "greater than" (Hanna et al., 2023) and basic arithmetic operations (Stolfo et al., 2023). Other work has studied how simple models solve modular arithmetic tasks (Nanda et al., 2023a). More behavioral approaches have also been explored. Tian et al. (2023) looked at how numbers generated by models can be used to test for answer calibration. Likewise, McCoy et al. (2024a; 2024b) showed how a Bayesian framework can be used to characterize how LLM performance on various tasks, including numeric ones, depends on the probability of the input, task, and output.

## 2.2 Probing and Representation Analysis in LLMs

Probing is a well-established technique for studying the internal representations of machine learning models (Alain & Bengio, 2018; Belinkov, 2021). Dating back to BERT (Devlin et al., 2019), probing has been used to explore various aspects of representation such as how language models encode sentence structure (Manning et al., 2020; Rogers et al., 2021; Tenney et al., 2019). More recently, probing has become a common technique in studying larger language models. In the work by Zhu et al. (2025) mentioned earlier, the authors designed linear probes to extract number value from latent embeddings. Probes have also been used to discover how language models process concepts across layers (Gurnee et al., 2023), and for extracting uncertainty in language model generations (Kadavath et al., 2022). Similarly, probes, and latent analysis more broadly, have been used in the world models literature, e.g., to show that language models trained on Othello games develop an internal model of the game (Nanda et al., 2023b), and that they learn to represent space and time (Gurnee & Tegmark, 2024; Gurnee et al., 2023; Nylund et al., 2024). More recent approaches to studying representations in language models have also been proposed such as activation patching (Meng et al., 2022), sparse autoencoders (Cunningham et al., 2023; Gao et al., 2024), and distributed alignment search (Geiger et al., 2024).

## 2.3 Behavioral Analysis of Language Models

By combining the prompt comprehension capabilities of large language models with the wide range of paradigms available in the behavioral sciences, a growing line of work proposes new tools for characterizing behavior in LLMs (Hardy et al., 2023; Ku et al., 2026; Ying et al., 2025). For example, Marjieh et al. (2024b) used a suite of perceptual judgment tasks from psychophysics to characterize sensory knowledge in LLMs. Binz & Schulz (2023) leveraged paradigms from cognitive psychology to study how GPT-3 solves various tasks. Bai et al. (2025) used tools from social psychology to diagnose implicit racial and gender bias in LLMs. Webb et al. (2023) subjected LLMs to analogical tasks to study their abstract reasoning capacities. Liu et al. (2025) tested LLMs on various tasks where humans are known to overthink to determine when LLMs are likely to do the same.

Our work is unique in that it integrates the strengths of the behavioral approaches of cognitive science with the probing techniques of machine learning and concretely translates them into practical applications. Indeed, prior work on number representations in LLMs often relies on probing analysis that may not be possible for closed-source models. Likewise, prior work on cognitive-behavioral analysis of LLMs rarely integrates probing techniques. Our work shows concretely how the two approaches can be combined to gain practical insight into LLM behavior that can be generalized beyond our specific domain of number.

# 3 Probing Number Representations in LLMs with Similarity Judgments

Our approach builds on the paradigm of similarity judgments from cognitive science (Marjieh et al., 2023; 2024a; Shepard, 1962; 1980; Sucholutsky et al., 2025; Tenenbaum & Griffiths, 2001; Tversky, 1977; Tversky & Hutchinson, 1986), which is also closely related to representational similarity analysis (RSA) from neuroscience (Kriegeskorte et al., 2008). The benefit of this approach is that it allows one to gain insight into representations by studying behavioral judgments. This is useful for cognitive scientists as they have no direct access into the human mind, and since LLMs often pose a similar challenge to machine learning scientists, our goal is to show how this paradigm can be helpful in this setting too.

Given a domain of interest $\mathcal{D}$, a set of representative items from that domain $\{x_1, x_2, \ldots, x_N\} \subset \mathcal{D}$ (or 'stimuli'), and an agent $\mathcal{A}$ whose representation $\mathcal{M}(\mathcal{D})$ one would like to characterize, the paradigm proceeds by eliciting pairwise similarity judgments from $\mathcal{A}$ across all pairs of items ('How similar is the item $x_i$ to the item $x_j$?'). These judgments are then aggregated into a similarity matrix $s_{ij}$ which defines a notion of proximity on $\mathcal{D}$ that can be used to characterize $\mathcal{M}(\mathcal{D})$. Since the notion of similarity is by construction neutral, it leaves it to the agent to impose its own structure on $s_{ij}$.

In our case, the domain of interest is the set of non-negative integers $\{0, 1, 2, \ldots\}$, and the agent is the LLM in question. We consider a representative sample of modern models, namely, GPT-4o (Hurst et al., 2024), two variants of Llama-3.1 (8b and 70b) (Dubey et al., 2024), DeepSeek-V3 (Liu et al., 2024), Claude-3.5-Sonnet (Anthropic, 2024), and Mixtral-8x22B (Jiang et al., 2024) (see Appendix F.4 for additional experiments with Qwen-2.5-7b; Qwen et al., 2025). We then apply the similarity technique in a series of experiments that span different contexts as well as behavioral (prompt-level) and internal (embedding-level) analyses and examine how the observed patterns decompose in terms of theoretical string and numerical metrics. We detail those experiments in what follows.

## 4 Methodology

### 4.1 Tasks

Our experiments fall into three categories, all of which are associated with a certain behavioral prompt for eliciting LLM judgments over numerical quantities. The prompts are provided in Appendix A. In the first category, we elicited similarity judgments over all pairs of numbers in the range $0 - 999$ ('How similar are the two numbers on a scale of 0 (completely dissimilar) to 1 (completely similar)?'; see Appendix A). We then constructed symmetric similarity matrices $s_{ij}$ by averaging over both presentation orders (i.e., $s(x_i, x_j)$ and $s(x_j, x_i)$), and then evaluated the structure that emerged (see Evaluation Metrics below) and the way it is affected by context. We focused on the range $0 - 999$ because all integers within it are represented as unique tokens in the models considered in the following section, with the exception of Mixtral-8x22B- Instruct-V0.1. This choice minimizes any tokenization-specific differences. In the second category, we extended the similarity analysis for one of the models for which we had internal access (Llama-3.1-8b; see also Appendix F.4 for a replication with Qwen-2.5-7b) and probed the extent to which its internal token embeddings reflected the external behavior through a linear transformation (see Probing Language Models). Finally, in the third category we constructed a naturalistic decision scenario in which the model had to select from two available numerical quantities $q_1, q_2$ (a compound concentration in a test tube) the one that is most similar to a desired quantity $q_0$ (further details in Experiments and Appendix A). We constructed such triplets $(q_0, q_1, q_2)$ in a way that the answer would diverge depending on whether the quantities are treated as numbers or strings (see Constructing Close Triplets below) which then allowed us to probe the string-bias of models.

### 4.2 Models

We evaluated four open-source and two closed-source models (and an additional model in Appendix F.4). The open-source models consist of Llama-3.1-8b-Instruct, Llama-3.1-70b-Instruct, DeepSeek-V3, Mixtral-8x22B-Instruct-V0.1, and the closed source include GPT-4o and Claude-3.5-Sonnet. Henceforth, we will refer to the Llama and Mixtral models as Llama-3.1-8b, Llama-3.1-70b, and Mixtral-8x22B to improve readability. Since generating $1000 \times 1000$ similarities requires sampling 1,000,000 pairwise comparisons which costs around $160 for Claude-3.5-Sonnet, we limit the model evaluations across all contexts to one run at a temperature of zero to get the most likely answer across the different settings. For comparison, we include results from an additional run in the basic similarity context at a temperature of 0.7 in Appendix F.

### 4.3 Evaluation Metrics

To characterize the extent to which the derived similarity matrices $s_{ij}$ were string-like or number-like, we regressed their values against two theoretical distance measures, namely, the Levenshtein string edit-distance $d_{Lev}(a, b)$ (Levenshtein, 1966) and the psychological Log-Linear number distance $d_{Log}(a, b)$ (Piantadosi, 2016) (we also considered a simple linear $\ell_1$ distance $|a - b|$ but found that it led to worse results; Appendix

Figure 7). The Log-Linear distance captures the psychological phenomenon whereby humans represent larger numbers with less fidelity leading to a logarithmic sensitivity to magnitude, i.e. $x - y \to \log(x) - \log(y)$ (see Appendix B for definition). The Levenshtein distance, on the other hand, is defined as the minimum number of deletions, insertions, or substitutions required to transform one string to the other. This can be defined recursively (see Appendix B) and we used the Python package `Levenshtein`[1] to compute it. For example, the Levenshtein distance between the digits 200 and 100 is 1 because it takes a single substitution ($2 \to 1$) to transform one string into the other.

Given the two distance measures evaluated on the range of interest $d_{ij}^{(Lev)}$ and $d_{ij}^{(Log)}$, we transformed them into similarity scores $s_{ij}(d) = 1 - d_{ij}/\max\{d_{ij}\}$ and then fitted them to $s_{ij}$ (after z-scoring), i.e., $s_{ij} = \alpha + \beta s_{ij}^{(Lev)} + \gamma s_{ij}^{(Log)}$ using the `LinearRegression` method from `scikit-learn` (Pedregosa et al., 2011) and computed the coefficient of determination ($R^2$ or variance-explained) which we also repeated for the individual metrics separately. We will refer to the multivariate regression analysis using both distance measures as the "combined" model. To derive confidence intervals, we bootstrapped over the number pairs (since the models had zero temperature) with replacement and 1,000 repetitions.

## 4.4 Probing Language Models

When generating an output, a language model first tokenizes the input sentence into a sequence of tokens $x_1, ..., x_n \in V$ where $V$ is some vocabulary. Then during the forward pass, the model incrementally transforms the initial embeddings (extracted from a vocabulary embedding layer) through each layer. The internal representations at each layer — commonly referred to as *residuals* — capture evolving latent features. These residuals can be analyzed to probe specific properties of the input or model behavior.

In our case, we train an affine transformation for each layer within the model that takes in a specific latent $h_i^{(j)}$ for token $i$ at layer $j$, and then learns a vector $w$ and bias term $b$, such that $w \cdot h_i^{(j)} + b$ predicts a specific distance measure. Due to computational constraints, the probes are run only on the Llama-3.1-8b model in the main text (however, we provide an additional replication of our probing analysis with another open-source model, Qwen-2.5-7b, in Appendix F.4). We take the original similarity prompt and extract the residuals from the last token — in our case the ":" after "Result:" (see Appendix A) — and train two linear probes on them. One probe is trained to predict the Levenshtein distance between the two inputted numbers, the other the Log-Linear distance.

To train the probes, we sample 10,000 random pairs of numbers in the range 0-999, train on 9,500 pairs, and evaluate on 500 held out test pairs (that were not part of the training set to avoid data leakage) by computing the correlation between the predicted value and the ground truth. We ablate across all layers and select layer 25 for the final probe (see Appendix Figure 6 for a robustness analysis on the number of training points, showing that performance is maintained even when we train on smaller sets, as well as an ablation analysis of performance across layers justifying our layer choice). To illustrate the results of the probe, we then compute their predictions on the range 0-500, yielding $500 \times 500$ similarity matrices which we then visualize using multidimensional scaling (MDS; Shepard, 1980). MDS is a visualization technique that represents high-dimensional data in a lower-dimensional space, preserving the relative pairwise distances or dissimilarities as closely as possible (we restrict to the range 0-500 to ensure that the MDS plots are not too crowded for visualization purposes only). We computed MDS embeddings using `scikit-learn`.

## 4.5 Constructing Close Triplets

Our goal was to construct diagnostic triplets for the naturalistic decision scenario such that (i) the options are numerically close to each other but there is an unambiguous correct answer, and (ii) the options are very different in terms of their edit distance relative to the target. To do that, we constructed the target quantity $q_0$ by randomly sampling three digits from the set $\{2, \ldots, 9\}$ with replacement and combining them into a number (e.g., $3, 3, 1 \to 331$). Then, we constructed a Levenshtein-aligned option by subtracting 1 from the largest decimal entry (i.e., $331 \to 231$). Finally, we constructed a Log-aligned option by keeping the largest decimal entry from the target and randomly sampling two new integers from the range $\{1, \ldots, 9\}$ excluding

---

[1] https://rapidfuzz.github.io/Levenshtein/

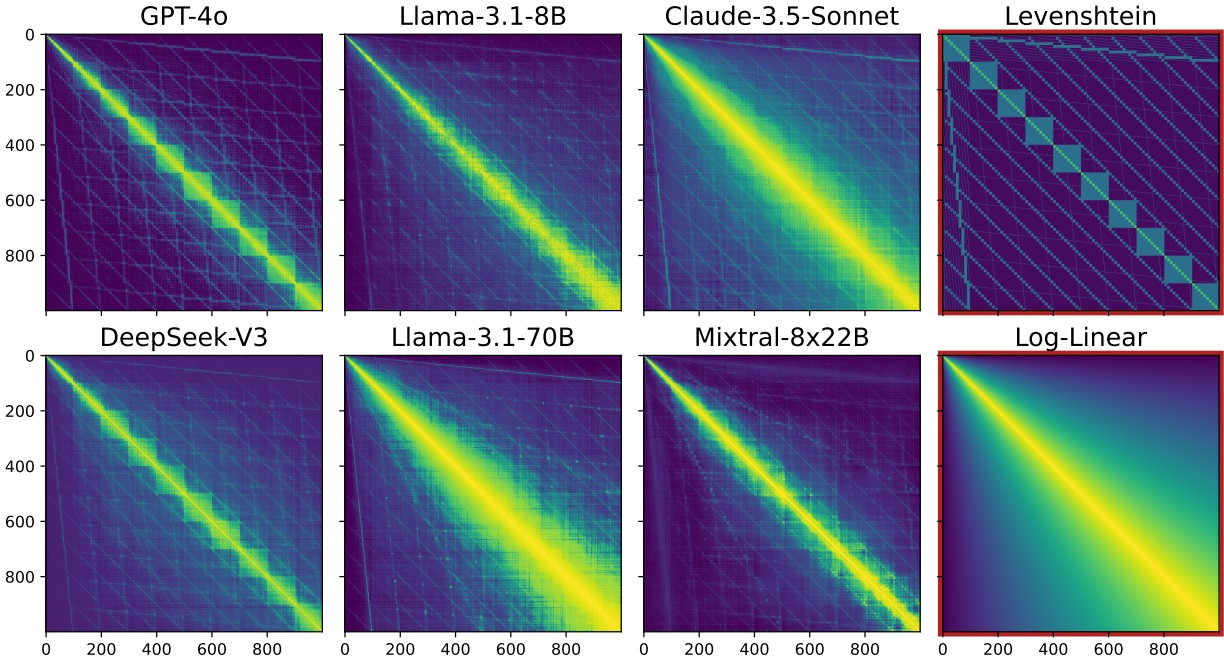

Figure 1: LLM number similarity matrices (symmetrized) over all integer pairs in the range $0 - 999$, along with two theoretical similarity matrices derived from a Levenshtein string edit distance and a psychological Log-Linear numerical distance (highlighted in dark red).

the digits that appeared in the other two numbers (e.g., $357$). Thus, the result would be $(331, 231, 357)$ in our example which satisfies the desired properties. We repeated the process also for five digit numbers to probe how things change for longer numbers (e.g., $25337, 15337, 26886$). We sampled 10,000 such triplets from each length category and took the unique subset. Overall, there were 6,474 unique three digit triplets, and 9,995 five digit triplets.

## 5 Experiments

### 5.1 Eliciting Similarity Judgments

In the first experiment, we elicited similarity judgments across all pairs of integers in the range $0 - 999$ using a suitable prompt ('How similar are the two numbers?'; see Appendix A for the full prompt; we reiterate that we used this range as it ensures that all integers are represented as unique tokens in our models, with the exception of Mixtral-8x22B, and thus controls for tokenization effects). We begin by presenting the qualitative patterns in the raw data and then quantify them using a suitable regression analysis. Figure 1 shows the similarity matrices for all models (see Methodology). The resulting LLM matrices are highly structured with a dominant area around the diagonal that in some cases (e.g., GPT-4o and DeepSeek-V3) takes a block-diagonal form, in addition to other sub-diagonal structures. These patterns appear to blend the properties found in two theoretical similarity matrices associated with string edit distance (Levenshtein) and numerical distance (Log-Linear) highlighted in the rightmost panels of Figure 1 (see Methodology). To further see whether these patterns arise from a tension between string and integer representations, in a following experiment we resolved the ambiguity of the similarity prompt by specifying the 'type' of the token using `int()` and `str()` contexts (see Appendix A for prompts). The results are shown in Figure 2A. In this case, we found that the context intervention pushed the matrices in opposing directions. For example, GPT-4o loses its block diagonal structure in the `int()` context, whereas Claude-3.5-Sonnet gains it in the `str()` context. Similar trends can also be found in the other models.

To quantify the above, we regressed the Levenshtein and Log-Linear distance measures against the LLM similarity judgments in the three contexts considered. A breakdown of the explained variance (coefficient of determination $R^2$) per condition are provided in Figure 2B. Overall, in the default context we found that a linear combination of the numerical and Levenshtein distance measures is sufficient to achieve an average $R^2$ of .726 (95% CI of mean: [.725, .726]), with the separate components each contributing significantly (Log-Linear only $R^2$ CI: [.607, .609], Levenshtein only $R^2$ CI: [.213, .215]). We note that replacing the Log-Linear distance with a simple linear $\ell_1$ distance diminishes combined average performance to an $R^2$ of .567 (95% CI: [.567, .568]; see Appendix Figure 7). In the other contexts, the metrics were correspondingly pushed in opposing directions. In the `str()` case, the mean $R^2$ CIs were: Combined: [.620, .621], Log-Linear: [.410, .412], and Levenshtein: [.309, .311]. On the other hand, in the `int()` context these were: Combined: [.721, .722], Log-Linear: [.645, .646], and Levenshtein: [.156, .158].

Next, we evaluated the extent to which the string-like similarity generalizes to less common number bases. The idea here is that if the model is relying on edit distance, then the effect should persist irrespective of how uncommon the underlying numerical basis is. To that end, we elicited additional similarity judgments for a subset of three models (GPT-4o, Llama-3.1-8b, and Llama-3.1-70b) using two additional number bases: base 4 and base 8 (see Methodology). The results are shown in Figure 3A. Here too we found highly structured matrices with patterns that are consistent with those derived from the Levenshtein distance. Quantitatively, we found that all bases resulted in a significant Levenshtein contribution (Figure 3B), and in the case of GPT-4o and Llama-3.1-70b those contributions were enhanced relative to the Log-Linear metric. Indeed, compare the $R^2$ CIs for the Log-Linear vs. Levenshtein regressors: In the case of Llama-3.1-70b, for base 10 we had: Log-Linear: [.780, .783] and Levenshtein: [.108, .112], whereas for base 4 this is reversed: Log-Linear: [.336, .340], and Levenshtein: [.366, .371]. Likewise, for GPT-4o in base 10: Log-Linear: [.426, .430], and Levenshtein: [.359, .364], whereas for base 4: Log-Linear: [.295, .299], and Levenshtein: [.424, .428].

## 5.2 Probing Internal Representations

The previous experiments showed that when an LLM is prompted for similarity judgments between two different integer tokens, the resulting behavioral metric is a combination of string and integer distance. In this section, we show how this also holds on the representational (i.e., token-embedding) level in a model for which we have internal access (Llama-3.1-8b). To do that, we train linear probes from the last token in the prompt to decode the Log-Linear and Levenshtein distances between the integers in question (see Methodology). This is non-trivial as there is no guarantee that such a linear transformation exists unless the model encodes the relevant information in its embedding. In Figure 4 we illustrate the decoded similarity matrices from the embeddings. Here we notice a pattern that is consistent with the behavioral (prompt-based) data. In the string probe case (right panel in Figure 4), the model is able to capture the edit-distance similarity between two numbers; whereas in the integer probe case (left panel in Figure 4), the decoded pattern is indeed much more log-linear as can be seen from the diagonal area in the matrix. Interestingly, however, string similarities still bleed into the representation as can be seen from the sub-diagonal structures. Quantitatively, the Pearson correlation between the string probe (on the held out data) and the Levenshtein and Log-Linear measures is .650 and .527, respectively. For the integer probe, the correlation is .917 with the Log-Linear distance and .393 with the Levenshtein distance (c.f., Table 2 in Appendix D).

To further highlight the structure of the latent subspaces, in Figure 4 we apply multidimensional scaling (Shepard, 1980) to the decoded similarity matrices in Figure 4 to derive two-dimensional maps of the relations between integers. We see that the probes nicely dissociate the integer and string subspaces. In the integer subspace, numbers are organized on a scale that captures numerical distance. On the other hand, in the string subspace, the scale is replaced with a non-linear pattern that closely follows edit distance. See Appendix F.4 for a replication of this probing analysis with another open-source model.

## 5.3 Behavioral Implications

In the final experiment we wanted to see whether there are behavioral implications for the number-string tension in a naturalistic scenario. To do that, we presented the models with an empirical setting in which they required a test tube with a certain unavailable compound concentration (given in parts per million

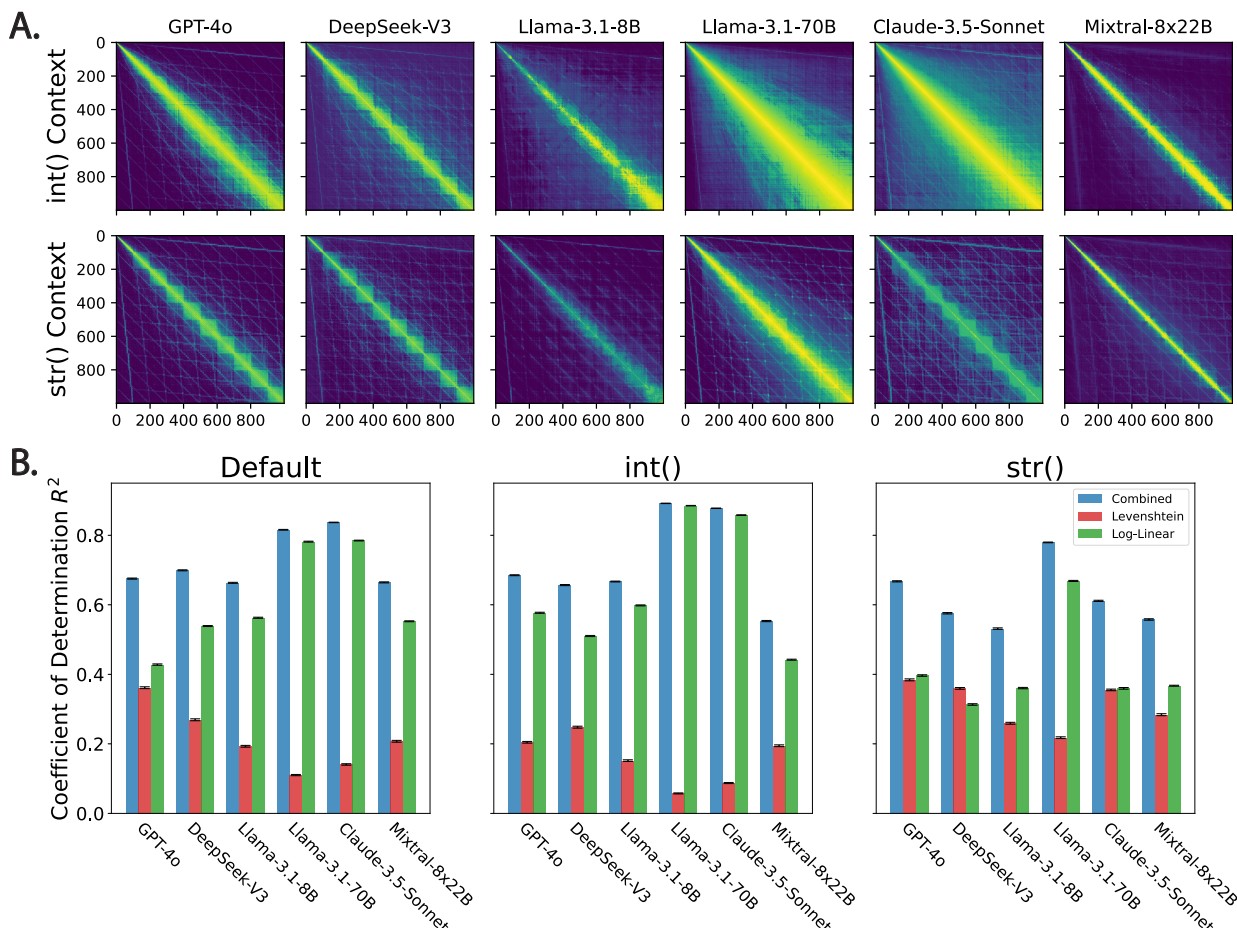

Figure 2: Context effects on LLM-elicited number similarity matrices and their decomposition. **A**. LLM similarity matrices under the effect of 'type' specification: `int()` vs. `str()` (see Appendix A for prompts). **B**. Coefficient of determination ($R^2$) for the different similarity matrices under the default (Figure 1), `int()`, and `str()` contexts for the combined (i.e., using both distance measures) and separate Levenshtein (string) and Log-Linear (numerical) distance predictors (error bars indicate 95% confidence intervals; see Methodology).

units; ppm). The models were asked to choose from two available test tubes the one containing the most similar concentration to the one desired. Crucially, we constructed diagnostic concentration triplets that lead to divergent answers depending on whether a Levenshtein distance is used or a Log-Linear one (see Methodology). Here is an example prompt: 'You require a compound with a concentration of approximately 785 ppm. Two test tubes are available: one containing 685 ppm and the other 791 ppm. Your task is to determine which test tube provides the most similar concentration to your required dosage.' Clearly, a concentration of 791 ppm is much more similar to 785 ppm than 685 ppm, but a model with strong string bias may erroneously choose 685 ppm which shares more digits with 785 ppm. We then evaluated the percentage of incorrect (Levenshtein-consistent) answers as a measure of string bias. In addition to the 3-digit triplets, we also considered 5-digit ones to see whether the problem may be exacerbated by the inclusion of more digits (e.g., 22565, 12565, and 28743; see Methodology). We also considered both orders of presentation to see whether there are any ordering effects when the Levenshtein-aligned option is presented first vs. the Log-Linear one ('Reverse' setting).

The results are shown in Figure 5. We found that all models had some degree of string-bias errors, but the range varied greatly and was enhanced for longer numbers. In the 3-digit case the largest percentage (averaged over order) was 36.86% for Llama-3.1-8b (followed by 11.38% for Llama-3.1-70b), and the smallest

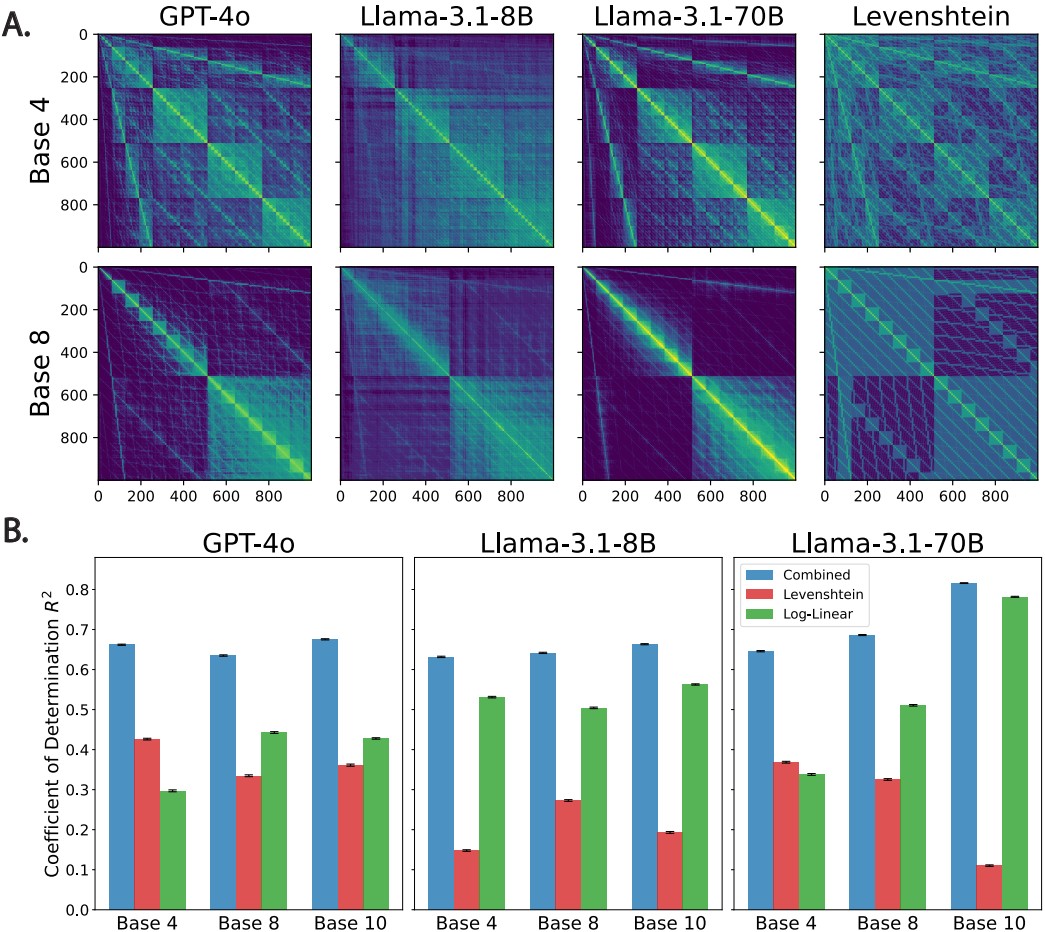

Figure 3: The effect of other number bases on elicited similarity. **A.** LLM similarity matrices over all integer pairs in the range $0 - 999$ represented in base 4 and 8 along with the corresponding Levenshtein distance measures (see Appendix A for prompts). **B.** Coefficient of determination ($R^2$) for the various similarity matrices under the different base contexts (including the base 10 results from Figure 1) for the combined and separate Levenshtein (string) and Log-Linear (numerical) distance predictors (error bars indicate 95% CIs).

was 0.02% for GPT-4o. Likewise, in the 5-digit case, the largest percentage was 47.03% for Llama-3.1-8b (followed by 19.04% for Llama-3.1-70b), and the smallest was 0.41% for GPT-4o. Interestingly, the Llama-3.1 models exhibited substantially larger degrees of bias, as well as some degree of order effects in some cases (note that this cannot be attributed to a generic presentation bias since in that case we wouldn't expect to see a difference between the 3-digit and 5-digit scenarios). Overall, these results suggest that the number-string tension can manifest itself in a realistic context, and that the extent that such a context can suppress the stringiness behavior varies across models.

## 6 Discussion

We have provided evidence for a representational tension in large language models that learn to represent numbers by processing text. This tension is reflected externally (through prompt-elicited similarity judgments), internally (through embedding probes), and also influences behavior in a quantitative decision scenario.

Our results suggest that the tension may arise from ambiguity in the 'type' of digit tokens in a given context. Specifically, by explicitly specifying the type of the tokens in the prompt (i.e., `int()` vs. `str()`) we showed how the similarity matrices in the default case can be pushed to align more strictly with string or log-linear

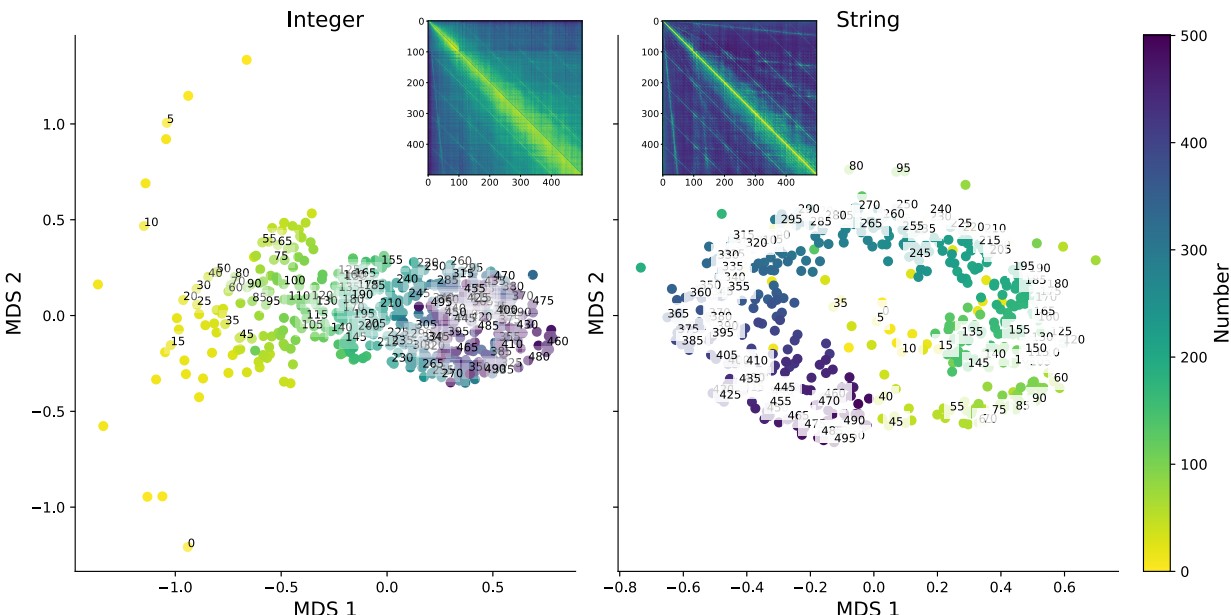

Figure 4: Decoded string and integer subspaces from Llama-3.1-8b using linear probes (see Methodology). The decoded similarity matrices are provided as insets along with their multidimensional scaling solutions (MDS). Integers are labeled every 5 points. See Appendix Figure 10 for a replication with another model.

distance. This ambiguity is reminiscent of polysemy and homonymy in language (Vicente & Falkum, 2017) whereby a single word can have multiple meanings, but in our case the ambiguity is extended across different systems. This entanglement of types is also reflected in our probing results whereby the linearly decoded similarity matrices had some residual shared structure.

Our findings are consistent with recent probing work showing that LLMs are able to encode the value of numbers (Zhu et al., 2025). Interestingly, however, we found that when the setting is not strictly mathematical as in arithmetic, LLMs appear to exhibit a more psychologically plausible log-linear representation, though blended with string distance. This is consistent with recent work showing that LLMs encode rich sensory knowledge (Marjieh et al., 2024b). The results also make contact with the literature on how LLMs perform numerical operations (Hanna et al., 2023; Stolfo et al., 2023) and the possible sources of numerical errors that could arise from their mode of learning (McCoy et al., 2024a;b). One possibility is that LLMs attempt to categorize the type of an integer token from context, and then use that to specify a set of operations on units of that type. For example, if an LLM views the numbers 1 and 2 as strings, then asking it to sum the two numbers may result in 12 rather than 3. Future work could interrogate whether there is indeed a causal link between such type-categorization mechanisms and numerical errors.

We end by discussing some limitations that point towards future directions. First, due to resource limitations, our LLM results were restricted to zero temperature (though see Appendix F), and the probing analysis was limited to smaller models. Future work could look into how the results generalize to a stochastic sampling setting and other larger models for probing.

Second, while we explored the effect of context on number similarity judgments, it remains to be seen whether one could causally intervene at the embedding level to steer the model from one token type to another.

Third, while our naturalistic decision scenario was designed to be as diagnostic as possible, further work is necessary to assess how pervasive this issue is in more generic daily numerical use cases. Relatedly, reasoning-based models are increasingly becoming the *de facto* standard in user-facing applications (e.g., OpenAI's o1, Jaech et al., 2024; DeepSeek's R1, Guo et al., 2025). Extending our findings to these models is a promising avenue for future work, as their ability to reason may fundamentally alter how similarity judgments are produced.

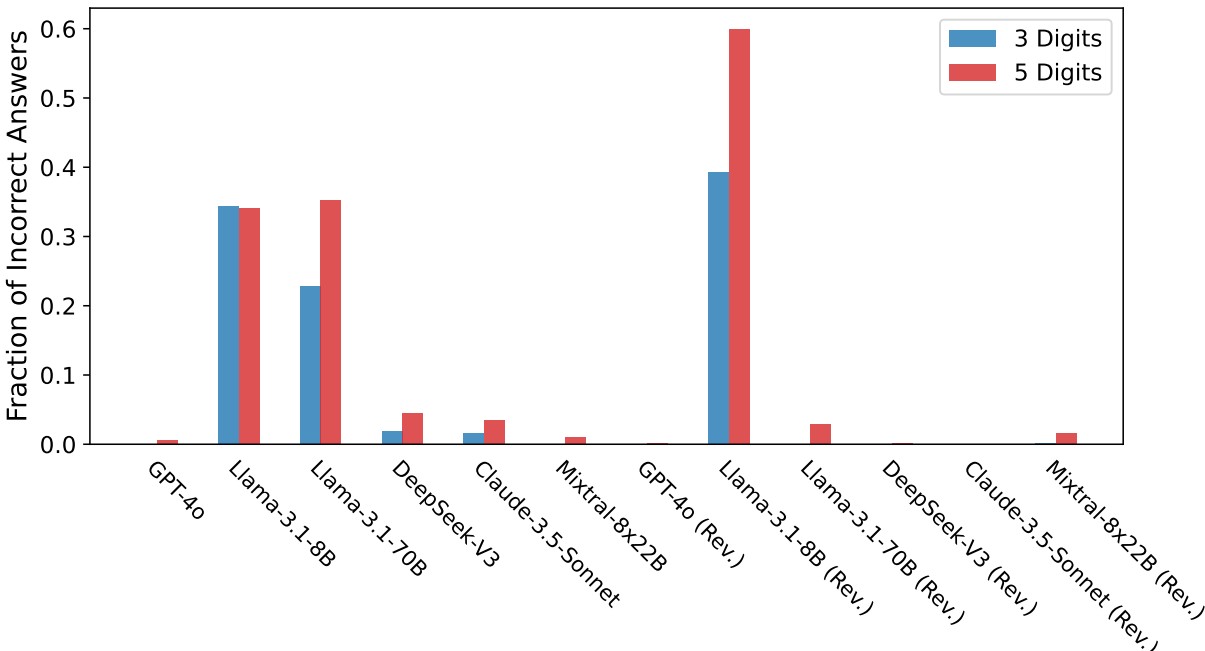

Figure 5: Probing string-bias in a naturalistic decision scenario. Bar plots indicate the fraction of times an incorrect (Levenshtein-aligned) option was chosen for the 3-digit and 5-digit scenarios considered, and the two possible presentation orders (see Methodology). '(Rev.)' indicates the case in which the Levenshtein-aligned option was presented second (see prompt in Appendix A).

Fourth, although it is tempting to connect the Levenshtein weight of the number similarity matrices directly to model error rates in the decision task, we found no significant correlation between them (this can also be intuitively seen, for instance, from the highly mixed integer-string GPT-4 similarity matrix in Figure 1 which does not strongly interfere with its performance on the decision task in Figure 5). This phenomenon is well known in cognitive science and has to do with the idea of representation-process pairs (Anderson, 1978; Nosofsky, 1992; Shepard, 1964). In a nutshell, when interpreting similarity judgments across tasks one must always consider the underlying representation in tandem with the process that operates on it: While the representation may be task-invariant, processes such as selective attention (Nosofsky, 1992; Shepard, 1964) can highlight different aspects of that representation in different tasks. Simple similarity judgment tasks such as rating the similarity of pairs of integers provide a neutral context (by design) that allows for different aspects of the underlying representation to surface. Finding evidence for a mixed integer-string similarity in that task implies that both aspects exist in the model representation, providing the experimenter with ideas on what other tasks to consider. However, there is no guarantee on how much these aspects will be activated in another independent task as that will depend on the process that operates in that task context. When considering a task such as our triplet decision scenario which provides a stronger context, different models will differ in their ability to suppress the string-like component (or equivalently, to focus their "attention" on the numerical component). It is possible that larger models are able to do that better in such familiar contexts (indeed, model size, as captured by log number of parameters, did correlate with model performance, $r = -.95, p = .05$; we excluded GPT-4o and Claude-3.5-Sonnet as their number of parameters is not publicly available). Future work could develop a more principled model of such selective attention processes which could be explored in tandem with additional probing work.

Finally, our work was restricted to the domain of numbers, but our results may be a special case of a broader symbol-string duality that impacts other domains (e.g., code and special characters). We hope to report on these directions in the future.

## 7 Conclusion

As artificial intelligence becomes increasingly intermingled with human intelligence, knowing what a number is becomes an increasingly important concern. McCulloch (1961) argued that the human capacity to understand numbers could be acquired by machines based on artificial neural networks. Our results suggest that there are still meaningful differences in how humans and large language models based on artificial neural networks conceive of numbers. More broadly, our work showcases the fruitful synergy between cognitive science and machine learning, demonstrating how human minds and opaque large AI systems pose similar interpretability challenges that can be addressed using behavioral paradigms such as similarity judgments.

## Broader Impact Statement

As noted throughout the paper, numbers are an essential part of how humans make sense of the world around them. As large language models become integrated in daily human decisions, it is important to understand how these models process and represent numbers to ensure that any potential misalignment (e.g., in a medical context) can be identified and mitigated. Our work directly taps into this important issue by providing new tools and insights into possible mechanisms.

## Acknowledgments

This work was supported by an Azure Foundation Models Research grant from Microsoft.

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

# A   Prompts

**Basic Similarity Prompt**

How similar are the two numbers on a scale of 0 (completely dissimilar) to 1 (completely similar)?
Respond only with the rating.
Number: {NUM1}
Number: {NUM2}
Rating:

**Integer Similarity Prompt**

How similar are the two numbers on a scale of 0 (completely dissimilar) to 1 (completely similar)?
Respond only with the rating.
Number: **int**({NUM1})
Number: **int**({NUM2})
Rating:

**String Similarity Prompt**

How similar are the two numbers on a scale of 0 (completely dissimilar) to 1 (completely similar)?
Respond only with the rating.
Number: **str**({NUM1})
Number: **str**({NUM2})
Rating:

**Different Base Similarity Prompt**

How similar are the two numbers on a scale of 0 (completely dissimilar) to 1 (completely similar)?
Respond only with the rating.
Base {BASE} number: {NUM1}
Base {BASE} number: {NUM2}
Rating:

**Compound Concentration Prompt**

You require a compound with a concentration of approximately {NUM1} ppm. Two test tubes are available: one containing {NUM2} ppm and the other {NUM3} ppm. Your task is to determine which test tube provides the most similar concentration to your required dosage. Which one will you choose? Respond only with the ppm value of the test tube you choose.

# B   Theoretical Distance Measures

We defined the Log-Linear psychological distance measure between two numbers $x$ and $y$ using the formula

$$d_{Log}(x, y) = 1 - \exp\left(-|\log(x + \epsilon) - \log(y + \epsilon)|\right)$$

where $\epsilon = 10^{-4}$ is a small regularizer to account for the fact that our domain included 0.

The Levenshtein edit distance $d_{Lev}(a, b)$ between two strings of characters $a = a_0 \ldots a_n$ and $b = b_0 \ldots b_n$ is defined recursively as

$$d_{Lev}(a,b) = \begin{cases} |a|, & \text{if} \quad |b| = 0 \\ |b|, & \text{if} \quad |a| = 0 \\ d_{Lev}(a_{1...n}, b_{1...n}), & \text{if} \quad a_0 = b_0 \\ 1 + \min\left\{ d_{Lev}(a_{1...n}, b), d_{Lev}(a, b_{1...n}), d_{Lev}(a_{1...n}, b_{1...n}) \right\}, & \text{else} \end{cases}$$

## C    Model versions

For reproducibility, we include the model versions used to collect data in Table 1, alongside the providers.

| Model Name | Model Version | Provider |
|---|---|---|
| Claude-3.5-Sonnet | Claude-3-5-Sonnet-20241022 | Claude API |
| DeepSeek-V3 | DeepSeek-V3 | together.ai |
| GPT-4o | GPT-4o-2024-08-06 | Azure API |
| Llama-3.1-8b | Llama-3.1-8b-Instruct-Turbo-128K | together.ai |
| Llama-3.1-70b | Llama-3.1-70b-Instruct-Turbo | together.ai |
| Mixtral-8x22B | Mixtral-8x22B-Instruct-V0.1 | together.ai |

Table 1: Full model versions alongside the model provider used to access them.

## D    Probing Analysis

### D.1    Background on Language Models

When a sentence $s$ is fed into a language model, the text is first tokenized into a sequence of tokens, $x_1, ..., x_t \in V$, where $V$ is the vocabulary of the model. Then when generating the $x_{t+1}$ token, the model, $f : \mathcal{X} \to \mathcal{Y}$ maps the input sequence to a probability distribution $\mathcal{Y} \in \mathbb{R}^{|V|}$. This is done by first embedding each of the input tokens $x_i$ using a learned input embedding matrix $E$ that maps the vocabulary to a set of embeddings denoted by $h_i^{(0)}$. As the token is processed throughout the model, each token's latent layer is updated as follows:

$$h_i^{(j)} = h_i^{(j-1)} + g(h_1^{(j-1)}, ..., h_i^{(j-1)})$$

Here $g$ is a function that typically consists of an MLP and an attention layer, alongside some form of normalization. Using the final representation of the last token $h_x^{(L)}$, an unembedding matrix is applied, $W$, which converts the latent vector back into the vocabulary space. We use the internal latents of the language model to train the probe.

### D.2    Training and Evaluation

The input data to the probe is the residuals of the last token from the different layers. In our case, this maps to the ":" after "Rating:". It is important to note that instruct models require specific prompts using roles to stay faithful to the training process. To account for this, we put the original task into the user message, and put the "Rating:" as the first tokens of the assistant message. We then continued generating from these forced tokens.

The probe itself consisted of a single affine layer predicting one value (the similarity score). In other words, by learning the probe we were essentially learning a mapping from the hidden dimension of the model (4096) to a one-dimensional space.

To train the probe, we first extract residuals from 10,000 random pairs of numbers between 0-999 across all 33 layers of Llama-3.1-8b (see also Appendix F.4). We then train probes on all layers and with varying dataset sizes (see below for the analysis). The groundtruth data is calculated by measuring either the Log-Linear or Levenshtein distance between all pairs of numbers. Then the probe is trained to predict these scores. The probe itself is trained with Adam for 100 epochs.

In Table 2 we report the correlation between the probes and the groundtruth Levenshtein and log-linear judgments.

|  | String Probe | Levenshtein | Int Probe | Log-Linear |
|---|---|---|---|---|
| String Probe | 1 | – | – | – |
| Levenshtein | 0.650*** | 1 | – | – |
| Int Probe | 0.667*** | 0.393*** | 1 | – |
| Log-Linear | 0.527*** | 0.266*** | 0.917*** | 1 |

Table 2: Correlation between the probes and the groundtruth number similarities based on Levenshtein and Log-Linear distance. All correlations have $p < 0.00001$.

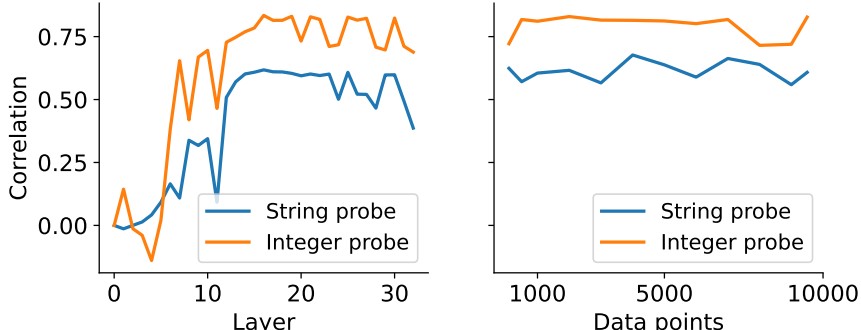

Figure 6: Ablations for the training of the probe. The vertical axis shows the correlation between the groundtruth similarities and the probe similarities on the test set. The plot on the left features how the correlation varies with the layer the probe is trained on using the full 9,500 data points for training. The right plot illustrates how the number of data points affects the correlation for a probe on layer 25.

# E   Linear Number Distance Control Analysis

As an additional control we reran our regression analysis with a linear $\ell_1$ number distance $|a - b|$ rather than the psychologically-motivated log linear one. The results are shown in Figure 7 and they yielded a worse fit relative to the Log-Linear case.

# F   Extensions

## F.1   Number of Comparisons

Within the main paper, we restricted the similarity judgments to a maximum of $1000 \times 1000$. Here we extend this for GPT-4o to the $2000 \times 2000$, presented in Figure 8. We find that the similarity pattern continues after the number 1000, but begins to get broader, possibly because of tokenization effects. To gain quantitative insight into this, we fitted the Log-Linear and Levenshtein metrics to the similarity matrix between the numbers in the range 1000-1999 and compared the results against our original ones in GPT-4o for the integers 0-999. In this case, we found that the average explained variance ($R^2$) of the combined model was .63, CI [.632, .636] (only Log-Linear: .54, CI [.539, .543], only Levenshtein: .23, CI [.230, .234]). For comparison, values for the range 0-999 were .67, CI [.674, .677] (only Log-Linear: .43, CI [.426, .430], only Levenshtein: .36, CI [.359, .364]). We see that both metrics continue to contribute meaningfully and in comparable magnitudes, however, more weight is placed on the Log-Linear metric as the block structure associated with string distance becomes less sharp in this range as shown in Figure 8.

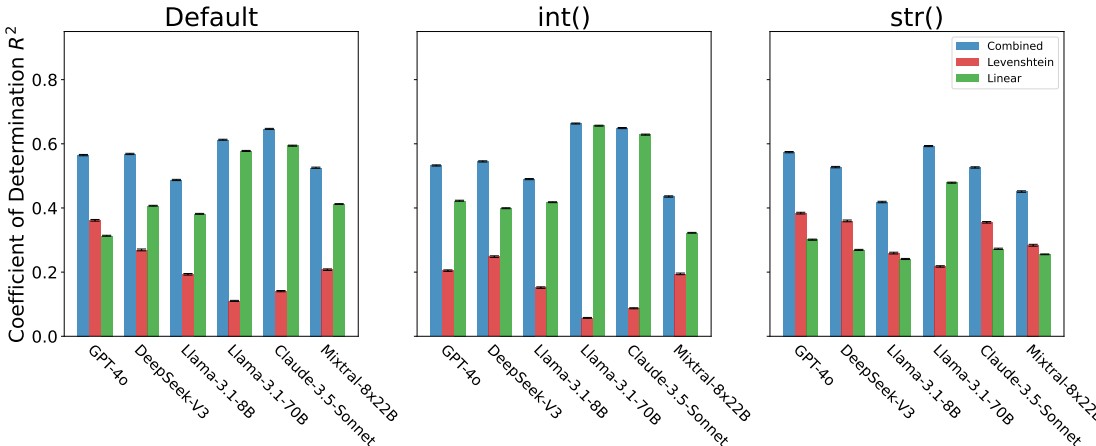

Figure 7: Coefficient of determination ($R^2$) for the different similarity matrices under the default (Figure 1), `int()`, and `str()` contexts for the combined and separate Levenshtein (string) and Linear (numerical) distance predictors (error bars indicate 95% confidence intervals; see Methodology).

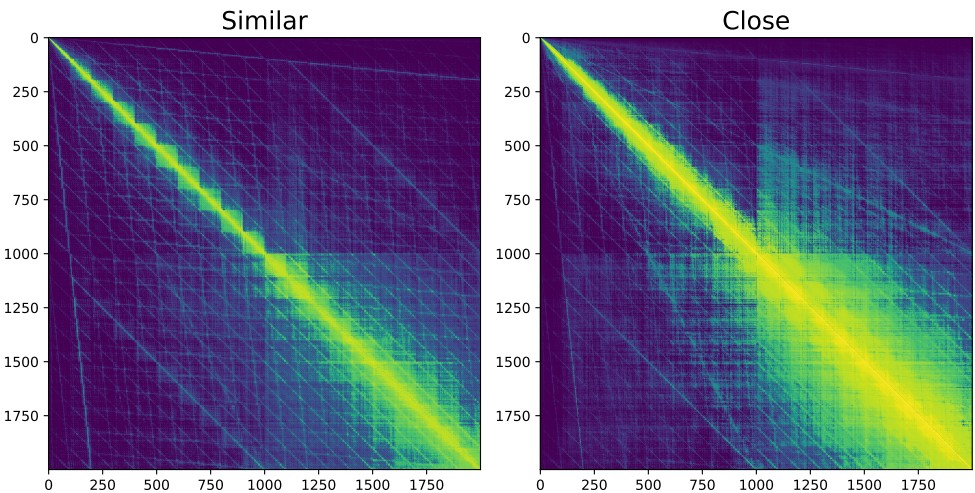

Figure 8: (left) Similarity judgments extended to the $2000 \times 2000$ setting. (right) Original prompt where "similar" is replaced with "closer" in the $2000 \times 2000$ setting. Both plots are for GPT-4o.

### F.2 Other Qualifiers

Throughout the paper, we used the "similar" qualifier when asking the model to compare two numbers. This is because "similar" is inherently neutral and is better motivated from the psychological literature for exploring representations. As an exploratory analysis we recomputed the GPT-4o similarity matrix using an alternative "close" qualifier which has a more quantitative connotation (similar to the `int()` type). We provide a side-by-side comparison in Figure 8. As expected, we observe that the "close" case has an effect that more closely aligns with the `int()` context. Nevertheless, we still notice a pronounced set of string-induced diagonals, especially as the integers get larger.

### F.3 Higher Temperature

We re-run the temperature 0 setting in the "basic similarity prompt" context at a temperature of 0.7. Using the new similarity matrices we measure the correlation and mean absolute difference across runs. We find

that across all models that correlation is above 0.787, and on average above 0.87. The model which had the lowest correlation was Mixtral-8x22B.

| Model Name | Correlation | Mean Absolute Difference |
|---|---|---|
| Claude-3.5-Sonnet | 0.956 | 0.0435 |
| DeepSeek-V3 | 0.864 | 0.0660 |
| GPT-4o | 0.862 | 0.0700 |
| Llama-3.1-8b | 0.831 | 0.1020 |
| Llama-3.1-70b | 0.916 | 0.0800 |
| Mixtral-8x22B | 0.787 | 0.0892 |

Table 3: Correlation and mean absolute distance between the "basic similarity prompt" with a temperature of 0 and a temperature of 0.7.

### F.4 Probing Analysis Replication with Another Open-Source Model (Qwen-2.5-7b)

We replicated our similarity and probing analysis with Qwen-2.5-7b (Qwen et al., 2025). The elicited prompt similarity matrix is shown in Figure 9A. We see that similar to the other models in Figure 1, the entangled integer-string pattern again emerges. This is further confirmed by rerunning the regression analysis with Levenshtein and Log-Linear distance predictors (Figure 9B) where the combined $R^2$ was .70, CI [.694, .700] (only Log-Linear .62, CI [.618, .621], only Levenshtein .17, CI [.163, .167]).

Turning next to the probing analysis, we sampled 100,000 random pairs of numbers in the range 0-999, trained on 9,500, and tested on the remaining 90,500 held-out pairs (note that this is a harder test than our original probing analysis). As with Llama-3.1-8b, our probes were able to reliably decode Log-Linear and Levenshtein distance from the model latents. In particular, the correlation for the integer probe (at the best layer 21) was .920 for Log-Linear (and .224 for Levenshtein), and for the string probe (at the best layer 18) was .618 for Levenshtein (and .378 for Log-Linear). This result can also be clearly seen by applying MDS to the decoded similarity values from each probe (Figure 10) where again two clear subspaces emerge corresponding to integer and string organization.

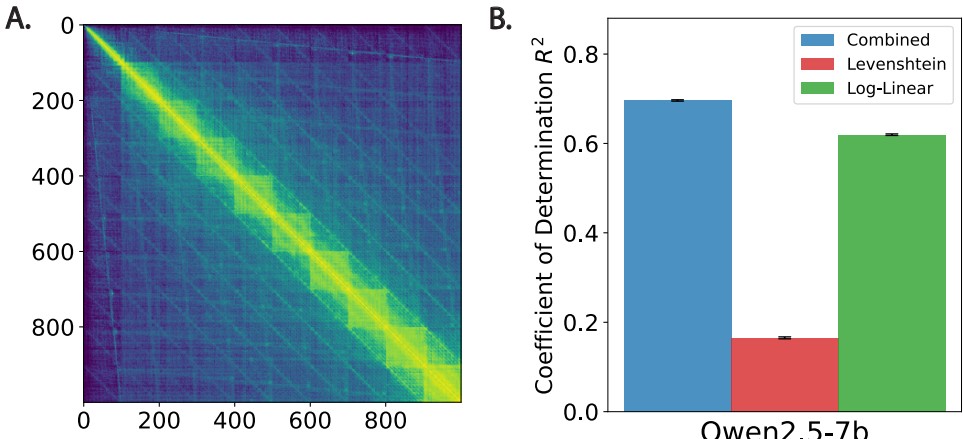

Figure 9: Additional open-source model control (Qwen-2.5-7b; Qwen et al., 2025). **A**. LLM number similarity matrix (symmetrized) for all integers in the range 0-999. **B**. Coefficient of determination ($R^2$) for the combined and separate Levenshtein (string) and Log-Linear (numerical) distance predictors (error bars are 95% CIs).

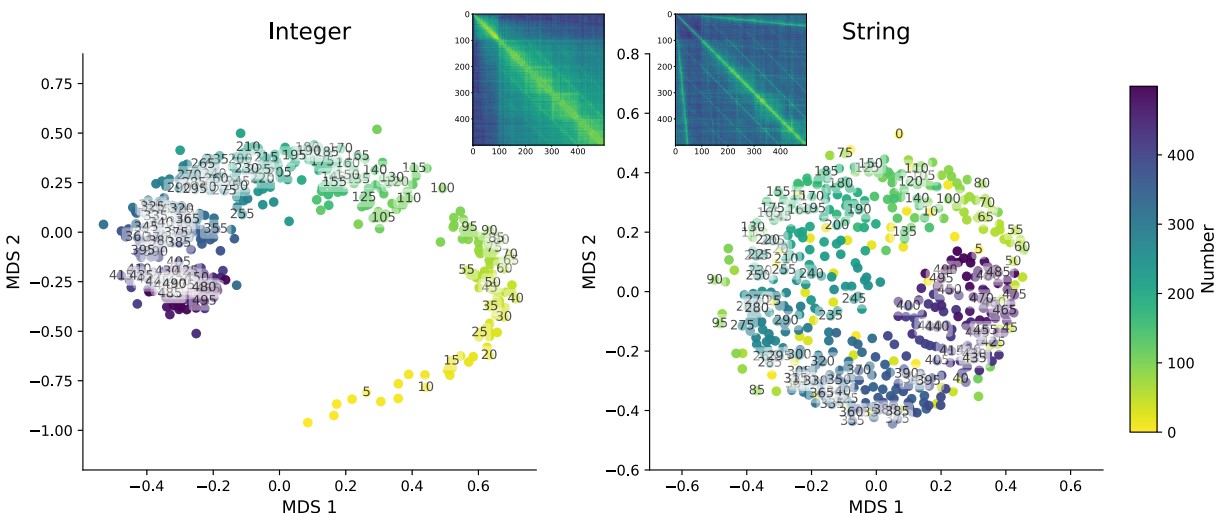

Figure 10: Decoded string and integer subspaces from Qwen-2.5-7b using linear probes (see Methodology). The decoded similarity matrices are provided as insets along with their multidimensional scaling solutions (MDS). Integers are labeled every 5 points.

