# OpenReview forum: "What is a Number, That a Large Language Model May Know It?"
_TMLR — Accepted by TMLR_

### Review · Reviewer_DMzW · 2026-03-13

**Summary Of Contributions:**

This paper investigates the long-standing problem of how large language models (LLMs) process and understand numbers. The same numerical value may be interpreted inconsistently due to different tokenization or context, which historically leads to undesired behaviors in computations and mathematical reasoning. The authors show that LLMs learn representational spaces that entangle string-like and numerical representations via analyses using a similarity-based prompting technique from cognitive science. The authors further show (1) how this entanglement is reflected in the latent embeddings, (2) how this entanglement can be reduced, but not entirely mitigated, through better engineering of the context, and (3) how this entanglement can propagate into a realistic decision scenario. These results collectively show the current status of LLM representations of numbers that might need collective efforts for future improvement.

Strengths
1. Figure 1 provides interesting insights. It qualitatively demonstrates the entanglement of string-like interpretation and numerical interpretation of numbers in LLMs, by showing that LLM number similarity matrices resemble both of the two theoretical similarity matrices, one derived from the Levenshtein string edit distance matrix and the other derived from the Log-Linear numerical distance matrix.
2. Figure 2 is also interesting. Upon reading Figure 1, I would like to suggest the authors to quantify the resemblance between the LLM number similarity matrices and the string-like versus number-like similarity matrices. This suggestion is directly addressed by Figure 2. In fact, the authors subsets the results by context, which makes the observation even more meaningful.
3. Overall, I think one contribution is the introduction of Levenshtein similarity matrix which can help explain a lot of patterns seen in the number similarity matrices of LLMs.

Weaknesses
1. While the figures contain rich information, the main text is not ordered in a way that facilitates reading and understanding. For example, Figure 2 appears on the top of page 4, but it’s only mentioned in the main text on page 7.
2. At first, I thought the analyses were performed by extracting the internal embeddings when the models parse certain numbers. After reading more carefully on the methodology, I realized that the results from Figures 1-3 were done in a prompt-based manner: the LLMs are explicitly asked to rate the similarity (on scale of 0 to 1) between two provided numbers. I would like the authors to (1) give a short justification on this prompt-based methodology, and (2) comment on why it seems to work well even if the prompt does not define the range of the numbers.

**Audience:**

Yes

**Audience Explanation:**

Understanding how LLMs process and understand numerical values is a longstanding area of research, and -- though this paper has not fully been there yet -- deciphering the failure modes of numerical comprehension can improve the performance on mathematical reasoning.

**Broader Impact Concerns:**

Nothing.

**Claims And Evidence:**

Yes

**Claims Explanation:**

The major claims are well supported by Figure 1 and Figure 2. The methodologies are reasonably well described.

**Requested Changes:**

1. Regarding Figure 1. The authors highlighted the borders of the two theoretical similarity matrices in black, but that color is very hard to distinguish from the dark color on the boundary of the matrices themselves. I barely noticed these two matrices are highlighted until I read the caption. I would recommend choosing a different color, such as firebrick.
2. **Question** regarding Figure 2: What does the “combined” mean? I do not seem to find the definition of it. Is it the results with respect to the average matrix between the Levenshtein-derived and Log-Linear-derived similarity matrices?

---

> ### Author Response · Authors · 2026-04-26
> **Response to Reviewer DMzW**
>
> Thank you for your evaluation and feedback! We detail our response to the raised points below.
>
> **3.1** _“While the figures contain rich information, the main text is not ordered in a way that facilitates reading and understanding. For example, Figure 2 appears on the top of page 4, but it’s only mentioned in the main text on page 7.”_
>
> **3.1 Response:** We updated the placement of our figures to improve readability. All figures now appear close to where they are mentioned.
>
> **3.2** _“At first, I thought the analyses were performed by extracting the internal embeddings when the models parse certain numbers. After reading more carefully on the methodology, I realized that the results from Figures 1-3 were done in a prompt-based manner: the LLMs are explicitly asked to rate the similarity (on scale of 0 to 1) between two provided numbers. I would like the authors to (1) give a short justification on this prompt-based methodology, and (2) comment on why it seems to work well even if the prompt does not define the range of the numbers.”_
>
> **3.2 Response:** As we note at the start of Section 3, our approach builds on a classic paradigm from cognitive science for characterizing representations using behavioral similarity judgments (Shepard, 1980; Tversky, 1977; Tenenbaum & Griffiths, 2001, Marjieh et al., 2024), which is also closely related to representational similarity analysis (RSA) from neuroscience
> (Kriegeskorte et al., 2008). This paradigm is useful because it allows us to gain insight into representations from behavioral judgments. Moreover, just as the internals of human minds are inaccessible to cognitive scientists, so are the internals of many modern LLMs to ML scientists, and thus similarity judgments provide a way to address this challenge. We now clarify this in the text. By further combining the prompt-based similarity analysis with probing (as seen in Figure 4 and Appendix Figure 10), we provide evidence that the patterns observed in the prompt-based similarity are also reflected in the internal representations of models.
>
> As for the second point, we believe that this follows from the fact that both the log-linear number distance and the Levenshtein string distance do not require knowing the full range of numbers to specify a degree of similarity (they only depend on the specific pair in question), and so if the model represents similarity in a way that is consistent with these distance measures (which is indeed the case given our empirical results) then we expect model responses to not require specifying the range of numbers.
>
> **3.3** _“Regarding Figure 1. The authors highlighted the borders of the two theoretical similarity matrices in black, but that color is very hard to distinguish from the dark color on the boundary of the matrices themselves. I barely noticed these two matrices are highlighted until I read the caption. I would recommend choosing a different color, such as firebrick.”_
>
> **3.3 Response:** Thank you for this suggestion, we updated the border color to firebrick.
>
> **3.4** _“Question regarding Figure 2: What does the “combined” mean? I do not seem to find the definition of it. Is it the results with respect to the average matrix between the Levenshtein-derived and Log-Linear-derived similarity matrices?”_
>
> **3.4 Response:** This corresponds to performing a regression analysis with both Levenshtein and Log-Linear distance as joint predictors of similarity (as opposed to only using one of them). We now clearly state that in the Methodology section, as well in the Figure caption.

---

### Review · Reviewer_a4vM · 2026-03-23

**Summary Of Contributions:**

This manuscript presents an analysis of the numeric representation learned by LLM's, concretely, comparing the latent spaces of numeric and string representation of numbers. The document shows that LLM's can, up to a certain degree, learn to map both spaces onto a common entangled latent space.

Authors were able to measure the discrepancy between latent representations coming from different input formats. And they also found that when an LLM is prompted for similarity judgments between two different integer tokens, the resulting behavioral metric is a combination of string and integer distances.

Strengths
- The introduction reads smoothly. It succeeds in introducing the context, the problem, and the proposed solution.
- The problem is of high relevance, given that it is known that LLM's might have troubles dealing with text representations of numbers, which leads to incorrect results when performing math operations.
- The analysis seems well designed, and the document suggests a correct experimentation protocol.

**Audience:**

Yes

**Audience Explanation:**

The problem is of high relevance to the community, and the findings of this work provide insights for better understanding of this problem.

**Broader Impact Concerns:**

None.

**Claims And Evidence:**

Yes

**Claims Explanation:**

The presentation is built upon the known fact that LLM's might have troubles dealing with text representations of numbers.

**Requested Changes:**

- The introduction should make more emphasis on the actual contribution of the paper.

- The related work section could be improved by making more emphasis on what makes this work different from previous ones.

- It is unclear why it was decided to train on numbers between 0 and 999, and evaluate on 0 to 500? Please justify the decision. Does it make any real difference having a split of numbers in this context? Moreover, the evaluation set includes the training set, which corresponds to a data leakage escenario from a machine learning point of view.

- Experiments and results are all mixed up, making it somehow unclear. Please, separate the experimental protocol design from the presentation of results.

- There is no section of conclusions. Or they are somehow mixed with the discussion. Please, organize them in a standard research paper outline.

- Many figures seem to appear at random positions, without previous citation of them in the main text. The common way to present objects in a text document, is to place them right after the paragraph of their first mention and always cite and explain them in the main text. Overall, avoiding that readers could bump into them without knowing how to interpret them.

---

> ### Author Response · Authors · 2026-04-26
> **Response to Reviewer a4vM (Part I)**
>
> Thank you! We appreciate the positive evaluation and the helpful feedback. Below we detail our response to the requested changes.
>
> **2.1** _“The introduction should make more emphasis on the actual contribution of the paper.”_
>
> **2.1 Response:** We added a summary of our contributions to the introduction:
>
> “To summarize, the contributions of our work are as follows:
> - We introduce a general technique from cognitive science based on similarity judgments for characterizing the representations of LLMs without having to access their internals.
> - We apply this technique to the domain of numbers, revealing representations that blend numerical and string components across six different models.
> - We show how similarity judgments can be combined with standard probing techniques to shed light on how prompt behavior is also reflected in model latents.
> - We show how this approach can be used to motivate additional experiments that reveal decision errors related to number understanding in LLMs.”
>
> **2.2** _“The related work section could be improved by making more emphasis on what makes this work different from previous ones.”_
>
> **2.2 Response:** We added the following clarification: “Our work is unique in that it integrates the strengths of the behavioral approaches of cognitive science with the probing techniques of machine learning and concretely translates them into practical applications. Indeed, prior work on number representations in LLMs often relies on probing analysis that may not be possible for closed-source models. Likewise, prior work on cognitive-behavioral analysis of LLMs rarely integrates probing techniques. Our work shows concretely how the two approaches can be combined to gain practical insight into LLM behavior that can be generalized beyond our specific domain of number.”
>
> **2.3** _“It is unclear why it was decided to train on numbers between 0 and 999, and evaluate on 0 to 500? Please justify the decision. Does it make any real difference having a split of numbers in this context? Moreover, the evaluation set includes the training set, which corresponds to a data leakage escenario from a machine learning point of view.”_
>
> **2.3 Response:** Apologies for the confusion as it may have been caused by a typo on our end in the main text (the information in the Appendix was reported correctly though). We updated the text to clarify that there was no data leakage in our analysis: “To train the probes, we sample 10,000 random pairs of numbers in the range 0-999, train on 9,500 pairs, and evaluate on 500 held out test pairs (that were not part of the training set to avoid data leakage) by computing the correlation between the predicted value and the ground truth. We ablate across all layers and select layer 25 for the final probe (see Appendix Figure 6 for a robustness analysis on the number of training points, showing that performance is maintained even when we train on smaller sets, as well as an ablation analysis of performance across layers justifying our layer choice). To illustrate the results of the probe, we then compute their predictions on the range 0-500, yielding 500×500 similarity matrices which we then visualize using multidimensional scaling (MDS; Shepard, 1980). MDS is a visualization technique that represents high-dimensional data in a lower-dimensional space, preserving the relative pairwise distances or dissimilarities as closely as possible (we restrict to the range 0-500 to ensure that the MDS plots are not too crowded for visualization purposes only).”
>
> As a control, in Appendix F.4 we replicate our results using an additional open-source model (Qwen-2.5-7b). In that case, we conducted a more conservative test by sampling 100,000 pairs in the range 0-999, training on 9,500 pairs, and testing on the remaining 90,500 held-out pairs. As with Llama-3.1-8b, our probes were able to reliably decode Log-Linear and Levenshtein distance from the model latents. In particular, the correlation for the integer probe (at the best layer 21) was .920 for Log-Linear (and .224 for Levenshtein), and for the string probe (at the best layer 18) was .618 for Levenshtein (and .378 for Log-Linear). Additional details and results are provided in Appendix F.4.

---

> > ### Author Response · Authors · 2026-04-26
> > **Response to Reviewer a4vM (Part II)**
> >
> > **2.4** _“Experiments and results are all mixed up, making it somehow unclear. Please, separate the experimental protocol design from the presentation of results.”_
> >
> > **2.4 Response:** We moved the results figures so that they are no longer mixed with the experimental design and so that they appear only after they are mentioned in the text.
> >
> > **2.5** _“There is no section of conclusions. Or they are somehow mixed with the discussion. Please, organize them in a standard research paper outline.”_
> >
> > **2.5 Response:** We now separate the conclusion section from the discussion.
> >
> > **2.6** _“Many figures seem to appear at random positions, without previous citation of them in the main text. The common way to present objects in a text document, is to place them right after the paragraph of their first mention and always cite and explain them in the main text. Overall, avoiding that readers could bump into them without knowing how to interpret them.”_
> >
> > **2.6 Response:** This has now been fixed following point 2.4.

---

> > > ### Comment · Reviewer_a4vM · 2026-04-28
> > > **Follow up on initial comments**
> > >
> > > *2.4* The experiments section has been improved for organization and consistency. Still, it would have been better to have separated sections for the experimental protocol and the results.
> > >
> > > *2.5* Discussion and conclusion have been separated now, as they should. Thank you. However, now it reads as if there is not so much to conclude. I am pretty sure that author can make an effort to have a more convincing conclusion section.
> > >
> > > *2.6* Thank you for improving the presentation and citation of Figures.

---

> > ### Comment · Reviewer_a4vM · 2026-04-28
> > **Agreed with these initial actions towards improving the manuscript**
> >
> > Thank you for clarifying those points, providing answers, and improving the manuscript. All answer provided are adequate.

---

### Review · Reviewer_vKWJ · 2026-03-25

**Summary Of Contributions:**

This paper investigates how large language models represent numbers, arguing that LLMs develop entangled representations that blend string-like and numerical properties. The authors elicit pairwise similarity judgments from six modern LLMs (GPT-4o, Claude-3.5-Sonnet, DeepSeek-V3, Llama-3.1-8B/70B, Mixtral-8x22B) over all integer pairs in [0, 999], and show that the resulting similarity matrices can be well-explained by a linear combination of Levenshtein edit distance (string metric) and Log-Linear psychological distance (numerical metric). The paper further demonstrates: (1) that explicit type-specifying contexts (int() vs. str()) shift but do not eliminate this entanglement; (2) that the same duality is recoverable from internal embeddings of Llama-3.1-8B via linear probes; (3) that this string-bias causes measurable errors in a naturalistic concentration-matching decision task, with Llama-3.1-8B reaching ~47% error rate on 5-digit triplets. The work is methodologically grounded in cognitive science's representational similarity analysis (RSA) tradition, enabling representation characterization without requiring model internals.

**Audience:**

Yes

**Audience Explanation:**

Yes. This paper sits at a productive intersection of mechanistic interpretability, cognitive science, and LLM behavioral analysis. The core finding (that LLMs treat numbers ambiguously as both strings and quantities) has broad relevance: it touches on arithmetic reliability, tokenization effects, representation learning, and practical failure modes. The cognitive science framing (RSA, Log-Linear psychological distance, similarity judgment paradigms) introduces methodological tools that are relatively novel to the ML community and could be widely adopted.

**Claims And Evidence:**

Yes

**Claims Explanation:**

Largely yes. The behavioral claim that LLM similarity judgments blend string and numerical distance is well-supported. The methodology is sound: symmetric similarity matrices are constructed from 1M pairwise queries, bootstrapped confidence intervals are reported, and a control with linear distance confirms the Log-Linear metric is the appropriate numerical baseline. The regression framework (R² decomposition) is straightforward and interpretable.

**Requested Changes:**

1. Extend probing to at least one additional model. The claim that string-number entanglement is present at the embedding level currently rests entirely on Llama-3.1-8B. Given that the behavioral results span six models, even a partial probing analysis on a larger model would significantly strengthen the claim.
2. Quantitatively connect similarity entanglement to decision errors. The paper currently presents Figure 2B (Levenshtein R² by model) and Figure 5 (error rates by model) independently. A scatter plot or correlation analysis linking a model's Levenshtein weight in the similarity decomposition to its error rate in the decision task would make the argument significantly more cohesive and causal.
3. Clarify the tokenization control more carefully. The authors state they restrict to 0–999 because these integers are unique tokens in all models considered. This should be verified explicitly for each model and reported in the main text or a table, as it is a critical assumption for the comparability of results.
4. The paper would benefit from a brief discussion of whether string-bias scales with tokenization granularity — numbers above 999 are often split into multiple tokens, and Figure 8 suggests the pattern "gets broader" above 1000. A more principled analysis of this boundary would be valuable.

---

> ### Author Response · Authors · 2026-04-26
> **Response to Reviewer vKWJ (Part I)**
>
> Thank you for your positive evaluation of our work! We detail our response to the requested changes below.
>
> **1.1** _“Extend probing to at least one additional model. The claim that string-number entanglement is present at the embedding level currently rests entirely on Llama-3.1-8B. Given that the behavioral results span six models, even a partial probing analysis on a larger model would significantly strengthen the claim.”_
>
> **1.1 Response:** We now include an additional replication of our similarity and probing analysis with another open-source model, Qwen-2.5-7b (we did not use Llama-3.1-70b or other larger models due to computational constraints). The results are summarized in Appendix F.4. Overall, we found that the prompt-elicited similarity matrix can again be effectively captured by a combination of log-linear and Levenshtein distance (Appendix Figure 9), achieving an $R^2$ of .70. Moreover, as with Llama-3.1-8b we were able to reliably decode the Log-Linear and Levenshtein distances from the model latents (.92 correlation between the integer probe and Log-Linear distance, and .62 correlation between the string probe and Levenshtein distance). Visualizing the decoded integer and string probe similarities using multi-dimensional scaling further revealed two subspaces with clear integer and string organization (Appendix Figure 10), consistent with the results of Figure 4.
>
> **1.2** _“Quantitatively connect similarity entanglement to decision errors. The paper currently presents Figure 2B (Levenshtein R² by model) and Figure 5 (error rates by model) independently. A scatter plot or correlation analysis linking a model's Levenshtein weight in the similarity decomposition to its error rate in the decision task would make the argument significantly more cohesive and causal.”_
>
> **1.2 Response:** Thank you for raising this interesting point. While it is tempting to connect the Levenshtein weight of the number similarity matrix directly to model error rate in the decision task, we did not find a significant correlation between them (this can also be intuitively seen, for instance, from the highly mixed integer-string GPT-4 similarity matrix in Figure 1 which does not strongly interfere with its performance on the decision task as seen in Figure 5). This phenomenon is well known in cognitive science and has to do with the idea of representation-process pairs (Nosofsky, 1992; Shepard, 1964; Anderson, 1978). In a nutshell, when interpreting similarity judgments across tasks one must always consider the underlying representation in tandem with the process that operates on it: While the representation may be task-invariant, processes such as selective attention (Nosofsky, 1992; Shepard, 1964) can highlight different aspects of that representation in different tasks. Simple similarity judgment tasks such as rating the similarity of pairs of integers provide a neutral context (by design) that allows for different aspects of the underlying representation to surface. Finding evidence for a mixed integer-string similarity in that task implies that both aspects exist in the model’s representation, providing the experimenter with ideas on what other tasks to consider. However, there is no guarantee on how much these aspects will be activated in another independent task as that will depend on the process that operates in that task context. When considering a task such as our triplet decision scenario which provides a stronger context, different models will differ in their ability to suppress the string-like component (or equivalently to focus their “attention” on the number component). It is possible that larger models are able to do that better in such familiar contexts (indeed, model size as captured by log number of parameters did correlate with model performance; r=-.95, p=.05; we excluded GPT-4o and Claude-Sonnet as the number of their parameters is not publicly available). Future work could develop a more principled model of such selective attention processes which could be explored in tandem with additional probing work. We now specifically state this in our discussion section.
>
> Nosofsky, R. M. (1992). Similarity scaling and cognitive process models. Annual review of Psychology, 43(1), 25-53.
>
> Shepard, R. N. (1964). Attention and the metric structure of the stimulus space. Journal of mathematical psychology, 1(1), 54-87.

---

> > ### Author Response · Authors · 2026-04-26
> > **Response to Reviewer vKWJ (Part II)**
> >
> > **1.3** _“Clarify the tokenization control more carefully. The authors state they restrict to 0–999 because these integers are unique tokens in all models considered. This should be verified explicitly for each model and reported in the main text or a table, as it is a critical assumption for the comparability of results.”_
> >
> > **1.3 Response:** We conducted an explicit check of tokenization and can confirm that all models reported in Figure 1 represent the integers 0-999 as unique tokens, with the exception of Mixtral-8x22B. However, as seen in Figures 1 and 2 in the paper Mixtral-8x22B behaved in a consistent way with the other models, suggesting that the effect of this difference is minor in this regime. We now explicitly state this fact in the text and nuance our statements, e.g. in Section 4.1: “We focused on the range 0 − 999 because all integers within it are represented as unique tokens in the models considered in the following section, with the exception of Mixtral-8x22B-Instruct-V0.1 . This choice minimizes any tokenization-specific differences.”
> >
> > **1.4** _“The paper would benefit from a brief discussion of whether string-bias scales with tokenization granularity — numbers above 999 are often split into multiple tokens, and Figure 8 suggests the pattern "gets broader" above 1000. A more principled analysis of this boundary would be valuable.”_
> >
> > **1.4 Response:** We now nuance our treatment in the paper: “To gain quantitative insight into this, we fitted the Log-Linear and Levenshtein metrics to the similarity matrix between the numbers in the range 1000-1999 and compared the results against our original ones in GPT-4o for the integers 0-999. In this case, we found that the average explained variance ($R^2$) of the combined model was .63, CI [.632, .636] (only Log-Linear: .54, CI [.539, .543], only Levenshtein: .23, CI [.230, .234]). For comparison, values for the range 0-999 were .67, CI [.674, .677] (only Log-Linear: .43, CI [.426, .430], only Levenshtein: .36, CI [.359, .364]). We see that both metrics continue to contribute meaningfully and in comparable magnitudes, however, more weight is placed on the Log-Linear metric as the block structure associated with string distance becomes less sharp in this range as shown in Figure 8.”
> >
> > Thank you again for your valuable feedback as it has helped us strengthen our work.

---

### Author Response · Authors · 2026-04-26
**Revision Summary**

We would like to thank all three reviewers for their favorable evaluations and helpful feedback. We have addressed all raised points in detail. In summary:

1. We now include a replication of our similarity and probing analysis using an additional open-source model (Qwen-2.5-7b) in a dedicated Appendix (F.4). The results are highly consistent with our probing analysis in Llama-3.1-8b.
2. We explored and improved additional analyses reported in the text following reviewer suggestions.
3. We clarified the contributions of our work and the way it differs from prior research.
4. We improved the organization of our figures and their integration in the text, in addition to other writing clarifications to enhance readability.

We believe that these edits have strengthened our work and would again like to thank the reviewers for their constructive suggestions!

---

### Decision · Action_Editor_zqzq · 2026-05-22

**Recommendation:** Accept as is

**Audience:**

Yes

**Audience Explanation:**

ML researchers who study number reasoning will find this useful. The link between cognitive science and LLM internals is generally interesting. The findings explain why models fail on simple number tasks. This work gives a clear path to better number representations.

**Claims And Evidence:**

Yes

**Claims Explanation:**

The experiments are clean and direct. Behavioral data and probing results match the theory. The authors test multiple models and find the same patterns. The evidence is solid and easy to verify.